**Data Availability Statement:** Data will be made available after acceptance.

**Funding:** Funding for the conduct of this study was provided by Abbott (www.abbott.com), Fundación

# Performance verification of the Abbott SARS-CoV-2 test for qualitative detection of IgG in Cali, Colombia

**Maria del Mar Castro**[1,2], **Isabella Caicedo**[3], **Helen Johanna Ortiz-Rojas**[3], **Carmen Manuela Castillo**[3], **Adriana Giovanna Medina**[3], **Neal Alexander**[1,2], **Maria Adelaida Gómez**[1,2]*, **Ludwig L. Albornoz**[3]*

**1** Centro Internacional de Entrenamiento e Investigaciones Médicas (CIDEIM), Cali, Colombia, **2** Universidad Icesi, Cali, Colombia, **3** Departamento de Patología y Medicina de Laboratorio, Fundación Valle del Lili, Cali, Colombia

* mgomez@cideim.org.co (MAG); ludwig.albornoz@fvl.org.co (LLA)

## Abstract

### Background

Adequate testing is critically important for control of the SARS-CoV-2 pandemic. Antibody testing is an option for case management and epidemiologic studies, with high specificity and variable sensitivity. However, characteristics of local populations may affect performance of these tests. For this reason, the National Institute of Health (INS) and regulatory agencies in Colombia require verification of diagnostic accuracy of tests introduced to the Colombian market.

### Methods

We conducted a validation study of the Abbott SARS-CoV-2 test for qualitative detection of IgG using the Abbott Architect i2000SR. Participants and retrospective samples were included from patients with suspected SARS-CoV-2 infection, age ≥18 years, and ≥8 days elapsed since initiation of symptoms. Pre-pandemic plasma samples (taken before October 2019) were used as controls. We estimated the sensitivity, specificity and agreement (kappa) of the Abbott IgG test compared to the gold standard (RT-PCR).

### Results

The overall sensitivity was 83.1% (95% CI: 75.4–100). Sensitivity among patients with ≥14 days since the start of symptoms was 85.7%, reaching 88% in samples collected from patients with COVID-19 symptoms onset >60 days. Specificity was 100% and the kappa index of agreement was 0.804 (95% CI: 0.642–0.965).

### Conclusions

Our findings show high sensitivity and specificity of the Abbott IgG test in a Colombian population, which meet the criteria set by the Colombian INS to aid in the diagnosis of COVID-19.

Restrepo-Barco (www.fundacionbarco.org) and
Fundación Valle del Lili (FVL, www.valledellili.org).
Test kits were provided by Abbott. The funders had
no role in study design, data collection and
analysis, decision to publish, or preparation of the
manuscript.

**Competing interests:** The study received funding
from Abbott (www.abbott.com), which also
provided the test kits. This does not alter our
adherence to PLOS ONE policies on sharing data
and materials. The authors have declared that no
competing interests exist.

Data from our patient groups also suggest that IgG response is detectable in a high proportion of individuals (88.1%) during the first two months following onset of symptoms.

## Introduction

Three highly pathogenic human coronaviruses have been identified to date, the Middle East respiratory syndrome coronavirus (MERS-CoV), the severe acute respiratory syndrome coronavirus (SARS-CoV) and the 2019 novel coronavirus (SARS-CoV-2). The coronavirus disease 2019 (COVID-19), caused by the infection with SARS-CoV-2 has caused millions of confirmed infections, resulting in more than 2.9 million fatalities worldwide [1]. The World Health Organization (WHO) has highlighted adequate testing as critically important for control of the pandemic [2]. Reverse transcription polymerase chain reaction (RT-PCR) is the current gold standard for the detection of SARS-CoV-2 [3] but has limitations related to cost and infrastructure, and its sensitivity changes with the duration of symptoms.

Serological tests play a decisive role in prevention, epidemiological surveillance and patient follow-up, especially for emerging infectious diseases [4, 5]. Antibody assay utility may be enhanced with strategies such as testing acute vs convalescent sera [6], or through orthogonal approaches [7–9]. The Cochrane Library published in June 2020 a review on the usefulness of antibody tests for identification of current and past SARS-CoV-2 infections [10]. Having analyzed 54 studies that used different antibody detection platforms, both commercial and in-house methods, the overall results showed a time-dependent increase in the sensitivity of antibody-based tests during active symptomatic infection, ranging from less than 30.1% in the first week for the pooled results of IgG, IgM, IgA total antibodies and IgG/IgM, to a pooled sensitivity of 91.4% (95% CI 87.0 to 94.4) during the third week of infection (days 15 to 21) for the combination of IgM/IgG.

In Colombia, dozens of different diagnostic tests based on the detection of IgM and / or IgG entered the market, prompting the development of minimal performance requirements by the Colombian National Institutes of Health (INS, Instituto Nacional de Salud) [11]. These consider that sensitivity and specificity may vary depending on factors such as the epidemiological and clinical settings, assay design and the viral genetics of SARS-CoV-2.

The Abbott SARS-CoV-2 test for qualitative detection of IgG on the Abbott Architect i2000SR system (hereafter referred as Abbott IgG test), is a chemiluminescent microparticle immunoassay that uses recombinant nucleocapsid protein for detection of anti-SARS-CoV-2 IgG antibodies in serum and plasma samples of patients with signs and symptoms of SARS-CoV-2 infection [12, 13]. According to its package insert, the intended use is as a diagnostic aid in detecting individuals with adaptive immunity against the virus, and claimed sensitivity of 100% and specificity of > 90% [12, 13]. Performance reports in different countries and communities, using the initial manufacturer recommended cutoff of 1.4, have consistently documented specificities >99%, and a range of sensitivities depending on the study population. Among some of these studies, sensitivity has been reported to be 68% in mild/moderate community managed cases in the United Kingdom [14], and 93.5% in UK patients with samples collected ≥ 14 days post symptoms onset [15]. Two independent studies in populations in the United States (US) reported 90.3% and 93.6% sensitivity in patients with symptoms onset between 15–21 days [16, 17], while 100% sensitivity was reported in a US cohort of RT-PCR positive patients, predominantly hospitalized, with samples obtained ≥17 days after onset of symptoms and ≥13 days after RT-PCR positivity [18].

In order to provide critical information for the use of serological tests in Colombia, we evaluated the performance, in terms of sensitivity and specificity, of the Abbott IgG test to aid in the diagnosis of COVID-19 in patients who consult at the Fundación Valle del Lili (FVL) University Hospital in Cali, Colombia. Our primary aim was to determine whether the Abbott IgG test achieved the goals of sensitivity ≥85% and specificity ≥90% for detection of SARS-CoV-2 infection, with reference to the gold standard of RT-PCR, as mandatory performance parameters defined by the Colombian INS. In addition, we sought to estimate the concordance between the performance of the Abbott IgG test and the gold standard of RT-PCR.

## Materials and methods

### Ethics statement

This research was approved and monitored by the Institutional Ethical Review Board (IRB) of Fundación Valle del Lili -FVL- (approval number: 286–2020) and Centro Internacional de Entrenamiento e Investigaciones Médicas -CIDEIM- (Approval number: 10–2020). Written informed consent was obtained from all prospectively recruited participants or their legal representatives, when unable to consent due to clinical reasons (e.g., ICU hospitalization). A waiver of consent for retrospective stored samples was approved by the IRBs. Medical decisions were not affected by participation in this study.

### Study design and population

We conducted a diagnostic secondary validation (verification) [11] study with prospective and retrospective participants. The prospective enrolment of participants was carried out at FVL, and retrospective (stored samples) were selected at CIDEIM and FVL in Cali, Colombia. Reporting of the study follows the Standard for the Reporting of Diagnostic Accuracy Studies (STARD 2015) guidelines (S1 Table) [19]. Two groups of participants were planned for enrolment in this study:

Group 1 (patients with suspected SARS-CoV-2 infection): inpatients and outpatients ≥18 years of age seeking healthcare in FVL, with suspected COVID-19 and ≥8 days after symptom onset, were invited to participate in the study (prospective participants). In addition, stored serum samples from patients with RT-PCR confirmed infection with SARS-CoV-2 were selected for inclusion in the study if relevant clinical information was available (retrospective participants). Exclusion criteria included immunosuppression (including HIV infection), autoimmune diseases and immunosuppressant drug treatment.

Group 2 (pre-pandemic controls): plasma samples obtained from EDTA anti-coagulated peripheral blood from patients consulting during January 2015 –September 2019, for unrelated pathologies (cutaneous ulcers), were selected from the biological specimen collection of CIDEIM. These corresponded to samples from patients ≥18 years of age, with diagnosis of cutaneous leishmaniasis (CL) and a negative HIV test.

Clinical samples: nasopharyngeal (NP) aspirate samples were obtained from all patients with suspected SARS-CoV-2 infection (i.e. Group 1), using 3mL sterile saline buffer. One 15 mL sample of venous blood was collected from each study participant with onset of symptoms ≥14 days. All serum samples were kept refrigerated until analysis (time to assay ranged between 1 and 6 hours after sample procurement).

### Index test: Abbott SARS-CoV-2 IgG test

This immunoassay is based on detection of IgG antibodies in plasma and serum samples, which bind to SARS-CoV-2 recombinant antigen coated microparticles, via a

chemiluminescent reaction measured by relative light units (RLU). The presence of SARS-CoV-2 IgG antibodies is determined by comparing the RLUs in the reaction containing the evaluated sample vs. the RLUs in the calibration control. The signal/cut off index (S/C) used for this study was 1.4 as recommended by the manufacturer (<1.4 negative, and ≥1.4 positive for anti-SARS-CoV-2 antibodies). Assay performance metrics met or exceeded manufacturer specifications as per product insert.

## Reference standard: SARS-CoV-2 RT-PCR

The RT-PCR assays were performed according to routine procedures at the FVL clinical laboratory. The SARS CoV-2 RT-PCR assays Allplex™ SARS-CoV-2 (Seegene, run on Bio-Rad CFX96 platform), AccuPower® SARS-CoV-2 Real-Time RT-PCR (Bioneer, run on Bioneer ExyStation™ 16 platform), GeneFinder™ COVID-19 Plus RealAmp (OSANG Healthcare, run on ELITe InGenius™ platform), BD SARS-CoV-2 (Becton, Dickinson, run on BD MAX™ platform), or VIASURE SARS-CoV-2 Real Time PCR™ (CerTest Biotec, run on QIAGEN Rotor-Gene-Q platform), were performed as reference standards. All RT-PCR assays complied with INS and FVL laboratory performance criteria. Only samples with a signal above the threshold in the relevant RT-PCR viral gene target regions for each assay were considered positive, as per the manufacturer instructions and internal standardized operating procedures (SOPs) at FVL. All assays and reagents were stored and handled following manufacturers' instructions. All operators performed the assays following the internal SOPs of FVL. Staff performing the index test were blinded to the results of RT-PCR while staff performing the reference standard were blinded to results of the index test.

## Clinical and laboratory data collection

Clinical and demographic data, including time since start of symptoms, disease severity and outcomes, history of autoimmune diseases and immunosuppression, were collected from the clinical records of patients or at the time of enrollment using a questionnaire. Laboratory results were entered into a case report form, sourced from the original laboratory information system records by a research assistant. All data were captured in a dedicated database utilizing Research Electronic Data Capture (REDCapTM, www.project-redcap.org), hosted at the FVL data center.

## Sample size

Based on the Colombian INS guidelines for performance verification studies for SARS-CoV-2 diagnostic tests [20, 21], a minimum of 16 symptomatic RT-PCR positive patient samples are required to meet regulatory criteria, considering goals of sensitivity ≥85% and specificity ≥90% for detection of SARS-CoV-2 infection, with reference to the gold standard of RT-PCR. For the power calculation, we used parameter values from the test insert [13]: 100% for sensitivity and 99.6% for specificity. We also assumed that study participants would be recruited prospectively from a group in which 9% would be symptomatic and RT-PCR positive. This group needed to number 178 in order to achieve the required sample size for sensitivity (9% of 178 is 16).

Time constraints imposed by the unfolding pandemic required this recruitment plan to be changed, and assessment of specificity was expedited by using a set of 61 banked pre-pandemic plasma samples as described above. Simulation of 100,000 repeated samples of this sample size, with 99.6% specificity, showed a power of 99.8% to establish specificity more than the 90% threshold, using the statistical methods described in the following section.

## Statistical analysis

For the primary objective, one-sided 95% confidence intervals are presented (since the hypotheses about sensitivity and specificity were one-sided, $\geq$85% y $\geq$90%, respectively), as well as two-sided 90% confidence intervals, using Wilson's method [22]. This analysis was repeated for subgroups defined in terms of time since the onset of symptoms. To determine agreement between the index test and the reference standard, as well as the crude agreement (proportion of cases in which the tests agree), the kappa index was calculated, the latter being a chance-adjusted measure of agreement [23]. Landis & Koch's descriptors for values of kappa were used [24]. Predictive values were calculated as functions of sensitivity, specificity and prevalence by standard identities [25]. Patients with invalid results or missing data on results of reference or index tests were excluded from analysis. Analysis was done using the R software, version 3.6.3.

## Results

Between September 17[th] and October 9[th], 2020, participants with presumed SARS-CoV-2 infections were invited to participate in this study. Overall, 260 potentially eligible retrospective participants were identified, of whom 48 had stored and available serum or plasma samples for IgG testing. In addition, 87 patients with RT-PCR confirmed SARS-CoV-2 infection were eligible to participate by donating a blood sample; among them, 49 were included in the study. Reasons for exclusion are detailed in Fig 1. No adverse events from performing the index test or the reference standard were reported by the specimen donors.

Most of the Group 1 participants (prospectively or retrospectively recruited patients with symptoms of COVID-19) were male (60.2%), with a mean age of 51.5 years (SD = 16.0). Fever, cough and difficulty in breathing were the most common symptoms (Table 1). Time from the initial symptoms until serologic testing was $\geq$ 14 days for 92.8% of participants, and > 60 days for 50.6% of participants. Median time between symptom initiation and respiratory specimen sampling for molecular testing was 7 days. Regarding the clinical presentation, most of the patients were classified as uncomplicated COVID-19 cases (35/83, 42.1%), followed by patients with severe disease (36.1%). Uncomplicated cases were defined as patients who did not consult the emergency room and were managed on an outpatient basis. Mild disease severity was considered as those patients that were hospitalized and did not require intensive care, and severe cases corresponded to patients who required intensive care. Four patients (4/83; 4.8%) had a fatal outcome.

For the pre-pandemic group, 61 eligible plasma samples from adult patients diagnosed with cutaneous leishmaniasis were identified in the biological collection of CIDEIM; of these, 59 met the quality standards for inclusion in the study. Most plasma samples (75%) were obtained from male participants with average age of 38 years (Fig 1).

### Sensitivity, specificity and agreement

The sensitivity of the index test (in Group 1), relative to RT-PCR, is shown in Table 2: overall it was 83.1%. The pre-specified subgroups, in terms of time elapsed since symptom onset, were 8–13 days and $\geq$14 days. A majority (93%) were in the latter category, and among these the sensitivity was 85.7%. Of samples collected from patients with onset of COVID-19 symptoms >60 days, 88% tested positive by the index test (Table 2), indicating that IgG responses can be tracked in a high proportion of individuals during the first two months following onset of symptoms.

The specificity of the test (in Group 2) was 100%, with all 59 samples testing negative on the index test. This means that the point estimate of the positive predictive value (PPV) is 100%.

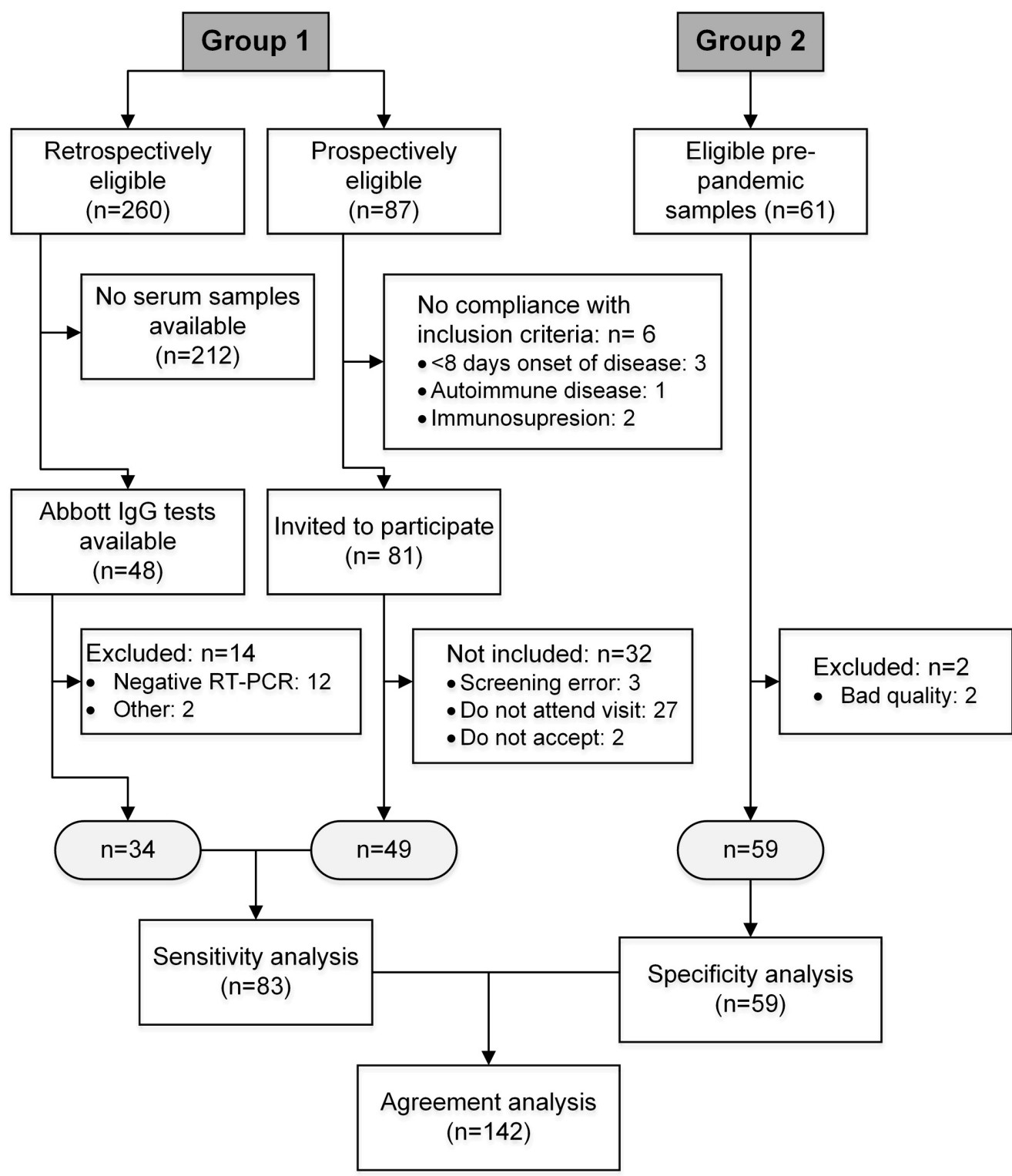

**Fig 1. Schematic representation of the study design and conduct.** Flow diagram of the performance verification study. Shown are all eligible and recruited patients and samples, as well as reasons for exclusion of participants during the study. Group 1: SARS-CoV-2 infected patients and samples. Group 2: Pre-pandemic samples.

**Table 1. Clinical and demographic characteristics of the study population.**

| Characteristic | Group 1 | | Total |
|---|---|---|---|
| | Included retrospectively | Included prospectively | |
| | n = 34 | n = 49 | n = 83 |
| Age (years): mean (SD) | 54.6 (17.2) | 49.4 (15.0) | 51.5 (16.0) |
| Female sex: n (%) | 13 (38.2) | 20 (40.8) | 33 (39.8) |
| Time since onset of symptoms | | | |
| Median (range) | 18 (10–58) | 85 (10–142) | 66 (10–142) |
| 8–13 days: n (%) | 5 (14.7) | 1 (2.0) | 6 (7.2) |
| ≥14 days: n (%) | 29 (85.3) | 48 (98.0) | 77 (92.8) |
| >60 days (subset of previous row): n (% of total) | 0 (0) | 42 (85.7) | 42 (50.6) |
| Disease severity: n (%) | | | |
| Uncomplicated | 12 (35.3) | 23 (46.9) | 35 (42.2) |
| Mild | 5 (14.7) | 13 (26.5) | 18 (21.7) |
| Severe | 17 (50) | 13 (26.5) | 30 (36.1) |
| Fatal outcome: n (%) | 4 (11.8) | 0 (0) | 4 (4.8) |
| Family history of autoimmune disease | | | |
| No | - | 35 (71.4) | 35 (42.2) |
| Yes | - | 8 (16.3) | 8 (9.6) |
| Missing | 34 (100) | 6 (12.2) | 40 (48.2) |
| Symptoms | | | |
| Fever: n (%) | 25 (73.5) | 26 (53.1) | 51 (61.4) |
| Expectoration: n (%) | 8 (23.5) | 3 (6.1) | 11 (13.3) |
| Asthenia o adynamia: n (%) | 19 (55.9) | 24 (49) | 43 (51.8) |
| Cough: n (%) | 27 (79.4) | 31 (63.3) | 58 (69.9) |
| Dyspnea: n (%) | 22 (64.7) | 19 (38.8) | 41 (49.4) |
| Anosmia: n (%) | 3 (8.8) | 8 (16.3) | 11 (13.3) |
| Myalgia: n (%) | 8 (23.5) | 19 (38.8) | 27 (32.5) |
| Arthralgia: n (%) | 8 (23.5) | 14 (28.6) | 22 (26.5) |
| Headache: n (%) | 4 (11.8) | 17 (34.7) | 21 (25.3) |
| Odynophagia: n (%) | 6 (17.6) | 20 (40.8) | 26 (31.3) |
| Gastrointestinal: n (%) | 6 (17.6) | 7 (14.3) | 13 (15.7) |

Fig 2 shows values of the negative predictive value (NPV), based on the point estimates of sensitivity and specificity from the current study, and various values of prevalence (i.e. positivity by RT-PCR). For a prevalence up to 33.3%, the NPV is over 90%.

As shown in Table 3, the kappa index of agreement was 0.804 (95% CI 0.642–0.965), which is in the range considered by Landis & Koch as "almost perfect" [24]. The crude agreement

**Table 2. Performance analysis.**

| | Index test result (Abbott IgG) | | |
|---|---|---|---|
| | Positive: n (sensitivity %, 95% one-sided CI, 90% two-sided CI) | Negative: n (%) | Total: n |
| Total | 69 (83.1) (75.4–100) (75.4–88.8) | 14 (16.9) | 83 |
| Onset of symptoms to sampling interval | | | |
| 8–13 days | 3 (50.0) (22.1–100) (22.1–77.9) | 3 (50.0) | 6 |
| ≥14 days | 66 (85.7) (77.9–100) (77.9–91.1) | 11 (14.3) | 77 |
| >60 days (subset of previous row) | 37 (88.1) (77.5–100) (77.5–94.1) | 5 (11.9) | 42 |

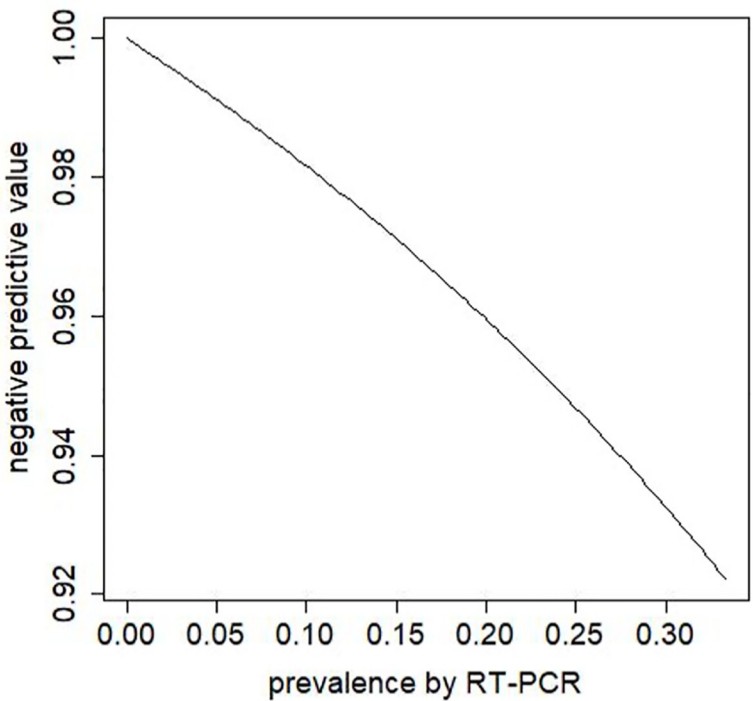

**Fig 2. Negative predictive value (NPV) of the Abbott IgG test.** NPVs were defined for the values of sensitivity and specificity in Table 2 (83 and 100% respectively), as a function of prevalence as determined by RT-PCR [25].

(percentage of results in agreement) was 90.1%. Among the 14 RT-PCR positive/IgG negative patients (Table 2), in three cases serum samples were obtained between 8–13 days after symptom onset.

## Post-hoc analyses: Exploring potential contributors to discrepancies between tests

As of October 2020, Abbott released technical information for implementation of a "grey zone" S/C index threshold ($\geq$0.49 and <1.4) in defining positive samples [26]. Applying this alternative threshold to the RT-PCR positive/IgG negative samples, 7 of 14 samples would be re-classified as IgG positive, while three of the 59 pre-pandemic samples would be re-classified as positive (with S/C indices of 0.50, 0.54 and 0.65).

The identity of the gene target for SARS-CoV-2 detection by RT-PCR was available for 80 of the 83 RT-PCR positive patients. Amplification of the envelope protein (E) and RNA-

**Table 3. Concordance analysis.**

| | Index test result (Abbott IgG) | | |
|---|---|---|---|
| | **Positive: n** | **Negative: n** | **Total: n (%)** |
| PCR positive (Group 1) | 69 | 14 | 83 (58.5) |
| PCR negative (Group 2) | 0 | 59 | 59 (41.5) |
| Total: n (%) | 69 (48.6) | 73 (51.4) | 142 |
| Index test result (Abbott IgG): subgroup analysis | | | |
| PCR positive (Group 1, interval since symptoms onset $\geq$60 days) | 37 | 5 | 42 (41.6) |
| PCR negative (Group 2) | 0 | 59 | 59 (58.4) |
| Total: n (%) | 37 (36.6) | 64 (63.4) | 101 |

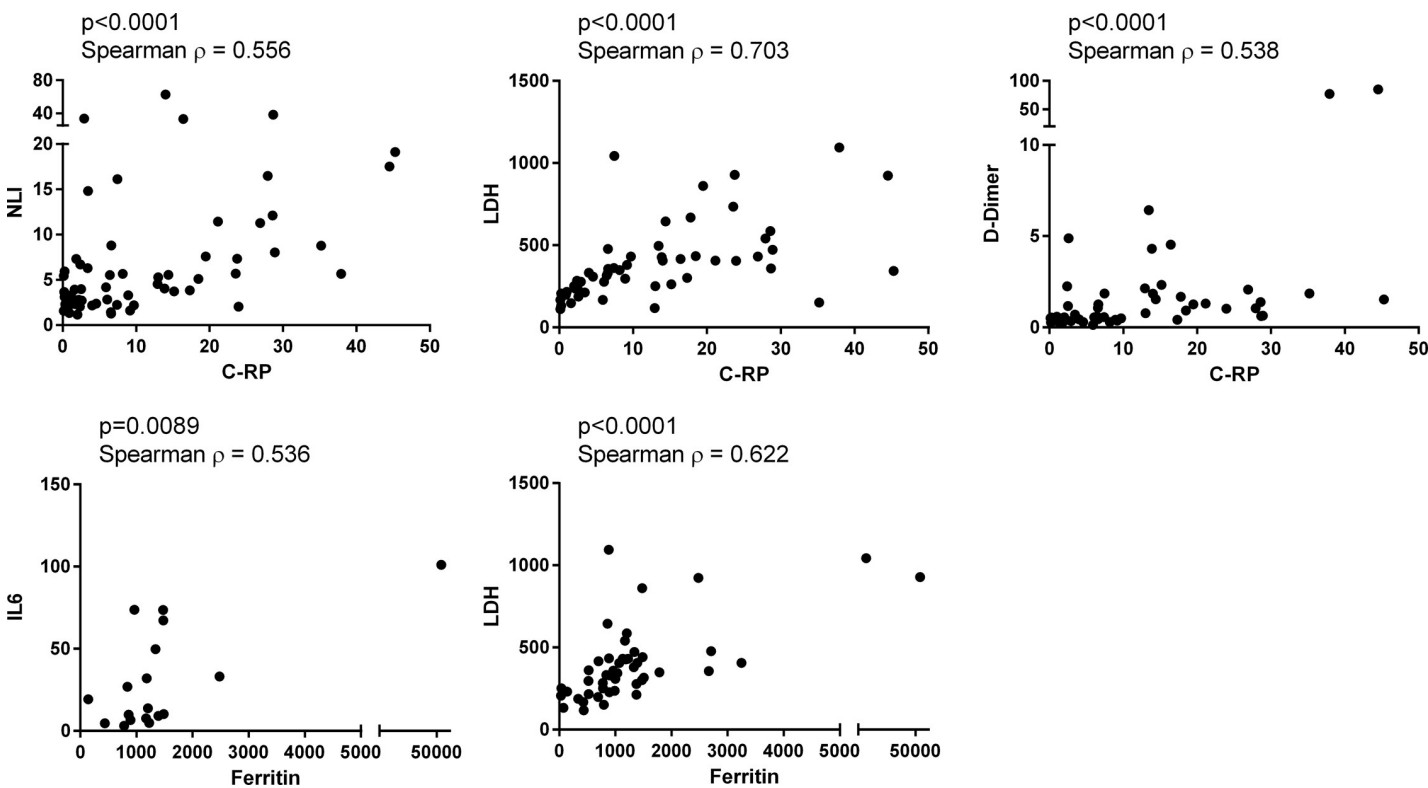

**Fig 3. Correlation of laboratory parameters of disease severity.** Data from laboratory tests was obtained from the clinical records of study participants. Shown are the graphical representation of the correlation analyses, as well as the parameters of significance and strength of the correlation for each data pair. C-RP (C-Reactive Protein), NLI (Neutrophil/Lymphocyte index), LDH (lactate dehydrogenase), D-Dimer (fibrin degradation product), IL6 (Interleukin 6).

dependent RNA polymerase (RdRp) genes was conducted in samples from 68 patients, while E, RdRp and nucleocapsid protein (N) genes were amplified in 7, N in 4 and ORF1ab in one sample. Ten of the 14 RT-PCR positive/IgG negative patients were diagnosed by amplification of E and RdRP targets. Ct values for at least one PCR target amplification product were available for 77 samples, and of these, 65 were processed with the AccuPower® SARS-CoV-2 Real-Time RT-PCR on a Bioneer ExyStation 16 platform. No correlation between Ct values and IgG test results were found with either E or RdRp targets.

We also explored possible relationships between Ct values, IgG S/C index and surrogates of disease severity (C-reactive protein -CRP-, D-dimer, lactate dehydrogenase -LDH-, ferritin, and interleukin 6 -IL6-), as possible contributors to the discrepant RT-PCR and Abbott IgG tests results, but no correlations were found. Sample quality was excluded as a possible contributor to discrepant RT-PCR and IgG results, since positive and significant correlations were found between different parameters of disease severity [27], within our patient cohort (Fig 3): CRP levels were positively and significantly correlated to the neutrophil/lymphocyte index (NLI), LDH and D-dimer ($p < 0.00001$, Spearman $\rho = 0.556$, $0.703$, $0.538$, respectively), while ferritin levels were positively and significantly correlated to LDH and IL6 ($p < 0.01$, Spearman $\rho = 0.622$, $0.536$, respectively).

## Discussion

With the increasing availability of anti-SARS-CoV-2 vaccines and the growing implementation of national vaccination strategies, quantitative and qualitative detection of anti-

SARS-CoV-2 antibodies has re-emerged as a central tool for epidemiological surveillance, in addition to supporting the diagnosis of infection. Here, we have verified the performance of the Abbott IgG test to aid in diagnosis of COVID-19 in adult patients consulting FVL, a high-level referral university hospital in Cali, Colombia. Our data show high sensitivity and specificity parameters when compared to the standard RT-PCR, indicating that Abbott IgG test meets the criteria set by the Colombian INS to aid in the diagnosis of COVID-19. It also meets the WHO criteria for specificity (≥97%) in tests aimed to detect prior SARS-CoV-2 infection [28]. Agreement with the molecular test was very good (kappa 0.804, 95% CI: 0.642–0.965), and the estimated sensitivity of the test remained above the threshold of ≥85% (as established by the Colombian guidelines for antibody testing), two months after infection, supporting its use in this population.

These results are similar to those reported in another study conducted by the Colombian INS, which showed sensitivity of 85.2% of the Abbott IgG test in RT-PCR positive symptomatic patients (n = 260), increasing to 97.2% in those patients with history of hospitalization due to COVID (n = 147) [20]. This study followed the protocol for secondary validation of the INS [11, 29], which sets a standard for the evaluation of diagnostic technologies for SARS-CoV-2 in the country and contributes to comparability of evaluations of tests across laboratories. Disease severity is a known factor related to the ability of current tests to detect antibody against SARS-CoV-2 [30]; in our study, most of the cases were mild and only 36% had severe disease, which may partly explain the estimated sensitivity in this study.

Another factor influencing the agreement between RT-PCR and serological tests is the time of sampling for each of the tests: samples obtained during the first 5–10 days after onset of symptoms are better suited for RT-PCR-based diagnosis as viral loads will likely be near their peak, while informative serum samples for IgG detection are best obtained after 14 days following onset of symptoms, in accordance with the time required for mounting an antibody response to SARS-CoV-2 infection [3, 31, 32]. In our cohort, among the 14 patients with RT-PCR positive/IgG negative results, only 3 corresponded to samples obtained between 8–13 days after onset of symptoms. Thus, other factors beyond time-to-sampling may contribute to this difference.

Based on the point estimates of sensitivity and specificity from our study, for a prevalence up to 33.3%, the NPV is over 90%. Colombia has recently completed the first nation-wide multicentric seroprevalence study of total (IgG/IgM) anti-SARS-CoV-2 antibodies, conducted between October and November, 2020 [33]. Estimated seropositivity frequencies ranged between 27% and 59% [33, 34], using the SARS-CoV-2 Total (COV2T) Advia Centaur–Siemens chemiluminescent immunoassay [34]. Considering the result of seroprevalence in Cali, which was 27% (CI 95%: 22–32%) [34], we estimate the negative predictive value of this test to be currently close to 93%.

Decline of antibody titers and seropositivity has been reported for different tests, including the Abbott IgG [30, 35–38]. In our study, sensitivity of the test in the subset of patients >60 days from symptom onset was 88%, indicating that IgG responses can be tracked in individuals during the first two months. This coincides with previous reports of a median half-life of antibodies detected by Abbott of 86 days [35], and a decay in seropositivity after 90 days reported in a Brazilian population evaluated with this test [39]. These aspects should be considered when analyzing and interpreting serosurveys conducted globally with the Abbott assay, since the true cumulative number of SARS-CoV-2 infections may be underestimated. However, knowledge of the test's performance and antibody dynamics would help to account for these aspects in estimation of seroprevalence [36, 40].

An important aspect of performance verification studies is the selection of the "reference standard". For antigen, as well as antibody-based diagnostic tools for COVID-19, the reference

standard has been defined as RT-PCR by national and international guidelines [29, 41]. However, a number of RT-PCR gene targets, commercial (and in-house) kits and amplification platforms are available for viral detection. Due to variability in the methodological conformation of these processes, these tools also differ in the amplification efficiencies and limits of detection (eg. lower limit of detection -LOD- or lower limit of quantitation–LLOQ-) which, as an example, can range from 3.8 to 23 copies/mL (LOD95) [42]. These differences introduce limitations that need to be considered for generalizability of the results and contrasts with other studies.

Ten of the 14 patients with discordant results were diagnosed by amplification of E and RdRP targets, while the Abbott IgG tests detects antibodies against the viral nucleocapsid protein. Differences in the molecular targets could contribute to the observed discrepancies and call for a more refined definition of what the reference standard for COVID-19 diagnostics is.

A limitation of our study was that separate groups of samples were used for the calculation of sensitivity and specificity, the latter coming from patients without respiratory symptoms, and lack of inclusion of asymptomatic patients which limits our ability to make inferences in this important group. Plasma and serum samples were used for this study, as indicated by the manufacturer. These samples are aligned with the acceptable characteristics of the WHO Target Product Profiles for antibody tests. However, samples that are the easier to collect (e.g. blood spots) can be developed as innovations to facilitate sampling and accessibility of the technology [28]. Presence of SARS-CoV-2 variants, and their potential effects on test performance, was not assessed in this study. Despite this, the population enrolled for testing was representative of the spectrum of patients and age groups seeking care in a reference facility in Colombia, with a breadth of clinical presentations (from mild to severely ill). Results from this study provide information on the specific use of this antibody test in the population for which it is intended to be used and highlight some important limitations in the interpretation of results based on the current reference standard definitions and duration of seropositivity.

## Supporting information

**S1 Table. STARD checklist.**
(DOCX)

## Acknowledgments

We gratefully acknowledge the volunteers who participated in this study and donated their sample. We thank the participation of Dr. Fernando Rosso and other members of the Clinical Research Center (CIC) at FVL for their support in patient recruitment, sample procurement and data management. Thanks to Johnnier Rojas Castrillon (FVL, CIC) for his support in patient recruitment and sample procurement. We also thank Oscar Oviedo (CIDEIM, statistics and epidemiology Unit), the data management team of the CIC at FVL for their support on generation and curation of databases, and Lina Giraldo (CIDEIM, Molecular Biology and Biochemistry Laboratory) for selection of pre-pandemic banked samples.

## Author Contributions

**Conceptualization:** Maria del Mar Castro, Neal Alexander, Maria Adelaida Gómez, Ludwig L. Albornoz.

**Data curation:** Neal Alexander.

**Formal analysis:** Neal Alexander.

**Funding acquisition:** Maria Adelaida Gómez, Ludwig L. Albornoz.

**Investigation:** Maria del Mar Castro, Isabella Caicedo, Helen Johanna Ortiz-Rojas, Carmen Manuela Castillo, Adriana Giovanna Medina.

**Methodology:** Maria del Mar Castro, Neal Alexander, Maria Adelaida Gómez, Ludwig L. Albornoz.

**Project administration:** Maria del Mar Castro, Isabella Caicedo.

**Resources:** Maria Adelaida Gómez, Ludwig L. Albornoz.

**Supervision:** Maria del Mar Castro, Maria Adelaida Gómez, Ludwig L. Albornoz.

**Validation:** Isabella Caicedo, Helen Johanna Ortiz-Rojas, Carmen Manuela Castillo, Adriana Giovanna Medina, Ludwig L. Albornoz.

**Visualization:** Maria del Mar Castro, Neal Alexander, Maria Adelaida Gómez.

**Writing – original draft:** Maria del Mar Castro, Maria Adelaida Gómez.

**Writing – review & editing:** Maria del Mar Castro, Isabella Caicedo, Helen Johanna Ortiz-Rojas, Carmen Manuela Castillo, Adriana Giovanna Medina, Neal Alexander, Maria Adelaida Gómez, Ludwig L. Albornoz.

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
