## [Decision Letter · Decision Letter 0]

31 May 2021

PONE-D-21-13284

Performance verification of the Abbott SARS-CoV-2 test for qualitative detection of IgG in Cali, Colombia.

PLOS ONE

Dear Dr. Gomez,

Thank you for submitting your manuscript to PLOS ONE. After careful consideration, we feel that it has merit but does not fully meet PLOS ONE’s publication criteria as it currently stands. Therefore, we invite you to submit a revised version of the manuscript that addresses the points raised during the review process.

We succeeded in obtaining the comments from only one reviewer, but failed to obtain them from the other. To progress a smooth review process, we determined to move to the next round for the review process. Please consider the serious comments from the reviewer.

We look forward to receiving your revised manuscript.

Kind regards,

Etsuro Ito

Academic Editor

PLOS ONE

Journal Requirements:

[Funding for the conduct of this study was provided by Abbott (www.abbott.com), Fundación Restrepo-Barco (www.fundacionbarco.org) and Fundación Valle del Lili (FVL, www.valledellili.org). Test kits were provided by Abbott.

The funders had no role in study design, data collection and analysis, decision to publish, or preparation of the manuscript.]. 

We note that you received funding from a commercial source: Abbott

Reviewers' comments:

Reviewer's Responses to Questions

**Comments to the Author**

1. Is the manuscript technically sound, and do the data support the conclusions?

Reviewer #1: Partly

2. Has the statistical analysis been performed appropriately and rigorously? 

Reviewer #1: Yes

3. Have the authors made all data underlying the findings in their manuscript fully available?

Reviewer #1: Yes

4. Is the manuscript presented in an intelligible fashion and written in standard English?

Reviewer #1: Yes

5. Review Comments to the Author

Reviewer #1: The study aimed to verify the diagnostic accuracy of serologic tests introduced to the Colombian market. Still, the analysis did not discuss the usual protocol that is used in Colombia. The researchers use serum o plasma samples for symptomatic SARS-CoV-2 positive patients to obtain the main conclusions. Still, the idea of viral diagnosis is to use blood drops without considering if the participant is symptomatic or asymptomatic to SARS-CoV-2 diagnosis with this IgG/IgM rapid tests development from Abbot company. Also, the INS study calculated SARS-CoV-2 seroprevalence in the general population in different regions in Colombia with this test; for example, to the city of Cali, seroprevalence was 30%. Is this percentage accurate?

It is important in the discussion that the researcher interprets the results at an epidemiological level. Follow-up epidemiological surveillance studies suggest that IgG detection in symptomatic and asymptomatic SARC-CoV-2 positive patients decreases in the following months. For this reason, some epidemiological studies suggest that it is not accurate to control the SARS-CoV-2 pandemic, neither case management nor epidemiologic surveillance. Also, at the virological level, understanding if a patient has been recently or previously infected with SARS-CoV-2 and how long antibodies stay in the body is an important next step in both detections of the virus and to decision of the interval period between vaccine doses.

6. PLOS authors have the option to publish the peer review history of their article (what does this mean?). If published, this will include your full peer review and any attached files.

Reviewer #1: No

---

## [Author Response · Author response to Decision Letter 0]

3 Aug 2021

Response to reviewers:

We thank the reviewers for their contributions towards improving the quality of this manuscript. Edits were made to the manuscript based on these recommendations and below we present a detailed response to their comments and questions:

Reviewer #1: The study aimed to verify the diagnostic accuracy of serologic tests introduced to the Colombian market. Still, the analysis did not discuss the usual protocol that is used in Colombia. 

R:// Thank you for this suggestion. The standard protocol for secondary validation of serologic tests of the Instituto Nacional de Salud de Colombia was followed throughout the study. We have added relevant aspects of the protocol to the discussion (lines 299-301).

The researchers use serum o plasma samples for symptomatic SARS-CoV-2 positive patients to obtain the main conclusions. Still, the idea of viral diagnosis is to use blood drops without considering if the participant is symptomatic or asymptomatic to SARS-CoV-2 diagnosis with this IgG/IgM rapid tests development from Abbot company. 

R:// Plasma and serum samples were used for the study, since these are the source samples to be used in these tests, as indicated by the manufacturer and test insert. Given the nature of this study, which is a secondary validation or verification study, we sought to replicate the conditions where the test will be used in the routine practice and the intended use of the manufacturer. This is in contrast to innovations on sampling strategies, which are relevant and part of the desirable characteristics of the WHO target product profiles for antibody tests, but out of the scope of our verification study. Edits were made to the discussion section for clarity (lines 325-334). Samples from both symptomatic (suspected SARS-CoV-2 infection) and pre-pandemic patients were included in the analysis. However, we did not include asymptomatic individuals with SARS-CoV-2 infection. This was acknowledged as a limitation of the study). 

Also, the INS study calculated SARS-CoV-2 seroprevalence in the general population in different regions in Colombia with this test; for example, to the city of Cali, seroprevalence was 30%. Is this percentage accurate? It is important in the discussion that the researcher interprets the results at an epidemiological level. Follow-up epidemiological surveillance studies suggest that IgG detection in symptomatic and asymptomatic SARC-CoV-2 positive patients decreases in the following months. For this reason, some epidemiological studies suggest that it is not accurate to control the SARS-CoV-2 pandemic, neither case management nor epidemiologic surveillance. Also, at the virological level, understanding if a patient has been recently or previously infected with SARS-CoV-2 and how long antibodies stay in the body is an important next step in both detections of the virus and to decision of the interval period between vaccine doses.

R:// The reviewer mentions a very important aspect of serologic testing for SARS-CoV-2. We have elaborated these aspects in the discussion section, considering new evidence generated in Colombia and elsewhere about the use of serologic testing for epidemiologic studies. These include the pre-print of the seroprevalence study in Colombia, which have been referenced and considered for the discussion. (Lines 316-334). The population of the current study was selected on the basis of evaluation of the diagnostic performance of the Abbott test in the hospital population where it is intended to be used (and not its utility as an epidemiological survey tool). Thus, most participants had suspected SARS-CoV-2 infection (Group 1) so are not representative of the general population (which includes e.g., uninfected, asymptomatic, symptomatic, vaccinated, etc). The lack of asymptomatic individuals in the study has been acknowledged as a limitation (lines 363 – 366).

---

## [Editor Report · Decision Letter 1]

10 Aug 2021

Performance verification of the Abbott SARS-CoV-2 test for qualitative detection of IgG in Cali, Colombia.

PONE-D-21-13284R1

Dear Dr. Gomez,

We’re pleased to inform you that your manuscript has been judged scientifically suitable for publication and will be formally accepted for publication once it meets all outstanding technical requirements.

Kind regards,

Etsuro Ito

Academic Editor

PLOS ONE

---

## [Editor Report · Acceptance letter]

23 Aug 2021

PONE-D-21-13284R1 

Performance verification of the Abbott SARS-CoV-2 test for qualitative detection of IgG in Cali, Colombia. 

Dear Dr. Gomez:

I'm pleased to inform you that your manuscript has been deemed suitable for publication in PLOS ONE. Congratulations! Your manuscript is now with our production department. 

Kind regards, 

on behalf of

Prof. Etsuro Ito 

Academic Editor

PLOS ONE